# Time-Frequency Domain Fusion Enhancement for Audio Super-Resolution

## ABSTRACT

Audio super-resolution aims to improve the quality of acoustic signals and is able to reconstruct corresponding high-resolution acoustic signals from low-resolution acoustic signals. However, since acoustic signals can be divided into two forms: time-domain acoustic waves or frequency-domain spectrograms, most existing research focuses on data enhancement in a single field, which can only obtain partial or local features of the audio signal, resulting in limitations of data analysis. Therefore, this paper proposes a time-frequency domain fusion enhanced audio super-resolution method to mine the complementarity of the two representations of acoustic signals. Specifically, we propose an end-to-end audio super-resolution network. Including the variational autoencoder based sound wave super-resolution module (SWSRM), U-Net-based Spectrogram Super-Resolution Module (SSRM), and attention-based Time-Frequency Domain Fusion Module (TFDFM). SWSRM and SSRM can generate more high-frequency and low-frequency components for audio respectively. As a critical component of our method, TFDFM performs weighted fusion on the above two outputs to obtain a super-resolution audio signal. Compared with other methods, experimental results on the VCTK and Piano datasets in natural scenes show that the time-frequency domain fusion audio super-resolution model has a state-of-the-art bandwidth expansion effect. Furthermore, we perform super-resolution on the ShipsEar dataset containing underwater acoustic signals. The super-resolution results are used to test ship target recognition, and and the accuracy is improved by 12.66%. Therefore, the proposed super-resolution method has excellent signal enhancement effect and generalization ability.

## KEYWORDS

Audio super-resolution, Time-frequency domain fusion, Sound wave, Spectrogram

## 1 INTRODUCTION

Audio is a digital representation of sound waves, which can be expressed in two ways: time domain and frequency domain. Audio processing tasks such as audio denoising[4], audio classification[25, 29], speaker recognition[21], and emotion recognition[30] use both forms of audio signals. Therefore, audio signal quality has an important impact on completing the above tasks. However, poor acoustic

sensor performance, attenuation during transmission, and over-compression of data can result in low-quality signals[26]. To obtain high-resolution (HR) audio, many audio super-resolution methods (SR) have been proposed[3, 7, 8, 11, 14, 17, 32]. Similar to image super-resolution, the purpose of audio super-resolution is to recover high-resolution audio from low-resolution(LR) audio.

Audio super-resolution has applications in many practical scenarios. For example, it can improve call quality in voice communication systems, improve music processing and audio editing, and improve the accuracy of speech recognition systems. Research shows that artificially expanding audio bandwidth can improve perception for people with hearing impairments[24]. This significantly improves the performance of speech enhancement [39], emotion recognition [12] and music reconstruction [38]. In addition, audio frequency signals are widely used in fields such as ocean exploration and underwater target detection because of their advantages such as long-distance transmission under water, penetrating obstacles, and good directionality. However, due to the influence of seawater salinity and temperature, ocean currents, and environmental noise, the quality of audio signals collected by sonar is often very poor. Through audio super-resolution, the quality of sonar signals can be effectively improved, thereby improving the accuracy of underwater target recognition and the accuracy of ocean detection. Therefore, audio super-resolution is required in various environments and transmission media.

In recent years, some researchers have utilized the time domain or frequency domain information of audio signals to expand the bandwidth to obtain high-quality audio. The time-domain super-resolution method acts directly on the original acoustic data, and theoretically can obtain speech input containing all original information, and the speech reconstruction process will not be limited by phase errors. However, due to its inherent high-dimensional long sequence structure, different time scales, and inability to directly obtain the frequency domain energy distribution, it has certain limitations in the deeper information processing of speech waveforms. The frequency domain super-resolution method can directly use the structured information in the low-frequency band of the spectrogram to construct the high-frequency spectrogram of the speech signal. However, such methods are highly dependent on the feature engineering of short-time Fourier transform, and the error in phase estimation also leads to large distortion when converting frequency domain signals into time domain signals. To sum up, using time domain or frequency domain data alone to train the network for bandwidth expansion can only capture some local features of cognitive objects, but cannot exploit the data correlation between different domains. This places limitations on the data analysis process, resulting in limited audio super-resolution effects. Therefore, dual-domain fusion audio super-resolution synthesis in time domain and frequency domain has great research value. Dual-domain fusion audio super-resolution aims to learn the relationship

between data in different domains and combine the comprehensive characteristics of the audio signal in the time domain and frequency domain to perform better super-resolution reconstruction of the audio signal.

In order to fully explore the relationship between the frequency domain and time domain of the sound signal to improve the super-resolution effect of the audio signal, in this paper, we propose a time-frequency domain fusion enhanced audio super-resolution method (TFDFE-ASR). The super-resolution model of this method consists of three parts: the VAE-based Sound Wave Super-Resolution Module (SWSRM), the U-Net-based Spectrogram Super-Resolution Module (SSRM) and Time-Frequency Domain Fusion Module(TFDFM) based on the attention feature pyramid. First, time-domain sound waves and frequency-domain spectrograms are defined as auditory and visual domain data, respectively. Then, SWSRM and SSRM are employed to perform audio super-resolution in the time domain and frequency domain respectively. The designed one-dimensional convolution and one-dimensional deconvolution layers of VAE can effectively enhance the high-frequency components of audio signal. And U-Net-based SSRM can enrich details of spectrogram by the multi-scale residual network and residual channel attention block. Finally, TFDFM is utilized to fuse the complementary information contained in the frequency and time domain data. Based on self-attention and time-frequency domain fusion attention, the common and unique characteristics of each data domain are retained as much as possible. Experimental results show that our method can be considered state-of-the-art compared to other methods, further enhancing the audio-visual effects on VCTK and Piano datasets. In addition, audio super-resolution effectively improves the accuracy of underwater acoustic signal target recognition.

## 2 RELATED WORK

Early audio super-resolution models use matrix decomposition for bandwidth expansion. However, training models can only be performed on small datasets due to the high computational cost of matrix decomposition [2]. Some audio encoders even directly truncate the high-frequency part of speech signal without encoding, and adopt speech enhancement to recover the high-frequency components. These methods do not take good advantage of the correlation between high and low-frequency signals. With the development of deep learning, convolutional neural networks (CNN) and generative adversarial networks (GAN) are widely used for multimedia processing tasks. The first convolutional architecture was proposed by Kuleshov et al. [15]. It is a bottleneck architecture model based on convolutional neural networks, using deep CNN to increase the sampling rate of the signal. Later on, many models used CNN to parse the audio. The audio signal is non-stationary, so CNN-based methods always fail to capture detailed information and produce over-smoothed results [22]. Hence, Eskimez et al. [6] proposed an audio super-resolution GAN to predict the high-frequency components of the log-power spectrum and reconstruct the time-domain signal. Li et al. [20] proposed a conditional GAN for audio super-resolution and achieved better performance than previous models. In recent years, it has emerged that attention mechanisms, instead of recurrent neural networks, can modulate the activation

of CNN [31]. It has shown better performance than traditional methods in computer vision.

In the current research, the phase of the high-frequency band is obtained by inverting the phase of the narrow band and adding a negative sign or by copying the phase of the narrow band spectrum. Most of the existing studies on audio super-resolution tasks have worked mainly in the frequency domain. Eskimez et al. [5] proposed an adversarial network structure using log-power spectrum (LPS) as input to generat the corresponding high-frequency LPS. This approach requires manual feature extraction and does not satisfy the end-to-end design. The conditional GAN model was used by Kumar et al. [16] to predict the missing high-frequency part in the amplitude spectrogram. Audio is generated by calculating the inverse Fourier transform using the full amplitude spectrogram and reusing the phase without changing the interpolated audio. Aaron et al. [28] proposed a WaveNet for generating original audio waveforms. The probability distribution of the current audio sample is predicted based on all the samples that have been generated before. This motivates some audio generation researchers to shift their focus to the time domain.

Some work explored audio super-resolution tasks directly in the time domain. For example, Ling et al. [23] applied hierarchical recurrent neural networks for audio super-resolution tasks. However, this method suffers from over-smoothing, where the model is unable to learn high-frequency data distributions and reconstruct detailed features in noisier speech segments with low signal-to-noise ratios. In [22], the time and frequency domains are combined for audio super-resolution. The authors proposed a time-frequency network utilizing supervision in both time and frequency domains. This model satisfied end-to-end training, and it is composed of a fully convolutional encoder-decoder network. The glow-based model solution was proposed by Zhang et al. [41] to encode low-resolution information in the time and frequency domains, respectively, using WaveNet and Glow integration. Su et al. [35] proposed BWE method is based on a feed-forward WaveNet architecture trained with a GAN-based deep feature loss. Different from the above audio super-resolution methods, our audio super-resolution model not only combines information in the time domain and frequency domain, but also analyzes the characteristics of the audio signal in the visual and auditory modes to better combine information from different domains, thereby outputting better super-resolution results.

## 3 METHODOLOGY

We consider that the TFDFE-ASR model requires both waveform and spectrogram, and design the SWSRM and SSRM as well as the attentional feature pyramid fusion module, respectively. The SWSRM is a SR network based on VAE for sound wave. VAE is a data enhancement method which is not affected by data format and has strong universality. In the paper, a series of frequency values sampled from LR audio act as input of VAE. Through multilayer convolution and deconvolution, the enhanced frequency vector is output by VAE, which includes more high-frequency component. We employ the reconstruction method based on Von-Mises-Distribution [36] to generate the SR sound wave combined with sampled phase and amplitude sequence.

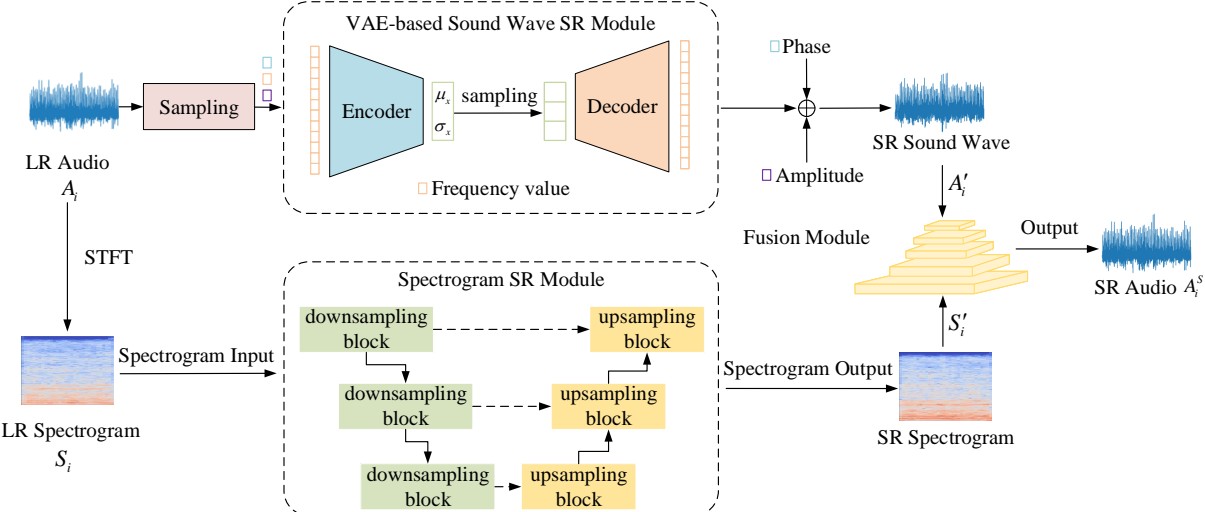

**Figure 1: Architecture illustration of TFDFE-ASR model, where SR is the abbreviation for super-resolution.**

The SSRM adopts the method of image super-resolution, which treats the spectrogram transformed by the acoustic signal as visual pattern. Therefore, the SSRM is design as a U-Net-based residual super-resolution method to generate the HR image (SR spectrogram). Finally, the SWSRM and the SSRM are combined using the attention-based pyramid fusion module to synthesize the HR audio. The TFDFE-ASR model is described in Figure 1, from which it can be seen that after the input of the LR audio $A_i$, the audio super-resolution reconstruction is performed by the VAE module to produce $A_i'$. The spectrogram $S_i$ obtained from the Fourier transform of the LR audio is used as the input of the U-Net dual regression module to perform the spectrogram SR reconstruction to time-frequency domain fusion module. The following subsections describe each part of TFDFE-ASR model, respectively.

## 3.1 Sound Wave Super-Resolution Module

When processed sound wave signal is not an integer period, the signal at the endpoints is obviously discontinuous. These discontinuous periodic signals can cause a number of high-frequency components, but they are not present in the original signal. Processing the discontinuous signals at endpoints by Fourier transform only results in some sort of artifact. This indicates that some high-frequency features, which can contribute significantly to the representation of the audio signal are an anomaly. For eliminating this misleading information, a windowing approach is used to generate a periodic signal before the Fourier transform on each audio frame and eliminate the contents located at the endpoints of each frame. The window size indicates the size of a segment of the audio signal, and the frame size is the number of samples considered in each signal block. This again introduces a problem in which some part of the signal is lost when the eliminated parts are joined together between adjacent frames. One effective solution to address this issue is by using overlapping frames.

As a acoustic representations, the waveform is used to train an end-to-end VAE with simple pre-processing (uniform sampling).

The frequency, phase and amplitude are uniformly sampled in the windowed LR audio signal. As the input of VAE, a series of frequency values are used for one-dimensional convolution and deconvolution. The amplitude and phase values sequence are adopted to recover the loudness of SR acoustic signal. The SWSRM employs VAE as it excels at representation learning without requiring any prior information, and learns the data automatically to ensure a low dimensional representation of the original data. The output of VAE combines the phase and amplitude sequence to synthesize the SR sound wave based on Von-Mises-Distribution [36].

**Module Architecture.** The architecture of the VAE is shown in Figure 2, which is an end-to-end audio-reconstructed model with an encoder and a decoder. Where the encoder serves to compress the frequency vector into a lower dimensional representation which is referred to as the latent space. The latent space is a representation of the raw data, only focusing on the most important features. However, the design of the potential space becomes extremely important to enable the encoder to compress the data efficiently and satisfy that the data has dependencies in different dimensions. In other words, if the different dimensions of the data are independent, it is essentially impossible to learn low-dimensional representations that capture the most important features. Each layer of the encoder in Figure 2 is composed of three parts: convolution/deconvolution, activation function, and batch normalization.

The convolutional kernels with 1×3, 1×5, or 1×7 in the convolutional layer can extract local features from subspaces of frequency vector and synthesize them to obtain global features. The same convolutional kernels can be used to process different input features or different parts of an input feature. The activation function enhances the characterization and the generalization of the model. ReLU is chosen as the activation function for the encoder part. Batch normalization is to transform the output of each layer into a distribution with a mean of 0 and a variance of 1. Maintaining parameter value within a reasonable interval can prevent gradient disappearance and explosion, and make the parameter updates faster. The

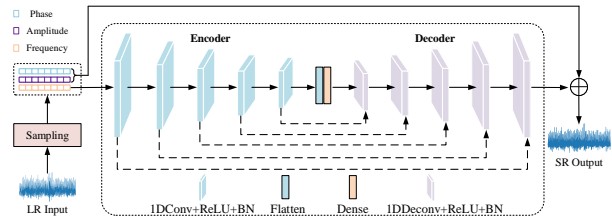

**Figure 2: VAE-based SWSRM, where LR and SR represent low-resolution and super-resolution, respectively.**

decoder restores the representation to the original domain, starting from the low-dimensional representation and trying to reconstruct the frequency vector. The backpropagation is used to minimize the reconstruction error. The reconstruction process expects the autoencoder to be sensitive enough to the input data and at the same time sufficiently vague not to memorize the original input to prevent over-fitting. The decoder compresses it using deconvolution, where each layer contains three parts: the deconvolution layer, the activation function, and the batch normalization. The latent space retains the most important features of the input data, creating a compact version of the original data. The encoder maps all the input vectors to a point in the potential space, and the decoder samples a new point in the potential space that is different from all the other points that have been learned. The decoder generates a meaningful new frequency vector.

The focus of VAE is on the design of the latent space, capturing the most important features and ignoring unnecessary details when encoding in the latent space. The audio signal generated by a low-dimensional latent space is more blurred; the audio signal generated by a high-dimensional latent space is less noisy and has a higher similarity with the original HR audio signal.

**Loss.** The loss function is a measure of the difference between true and predicted results during model training. The goal of the training process is also to minimize the loss function by comparing the distance between the original vector and the reconstructed vector element-wise using root mean square error (RMSE). Using KL loss provides the difference between the normal distribution and the standard normal distribution. By continuously changing the two parameters of the minimization vector and the variance vector, the KL loss again analyzes the difference between the normal distribution identified by the encoder and the standard normal distribution. The RMSE is defined as:

$$RMSE = \sqrt{\frac{1}{N} \sum_{t=1}^{N} \left(f_t - \hat{f}_t\right)^2} \qquad (1)$$

where $f_t$ and $\hat{f}_t$ denote the original vector and the reconstructed vector, respectively. $N$ is the number of sampling points.

The difference between the Gaussian distribution $N(\mu, \sigma^2)$ and the standard normal distribution $N(0, 1)$ is calculated, and this distance is used as the loss to regulate $N(\mu, \sigma^2)$ toward $N(0, 1)$. The KL loss can be defined as:

$$D_{KL}\left(N\left(\mu, \sigma^2\right) \| N(0, 1)\right) = \frac{1}{2} \sum \left(1 + \log\left(\sigma^2\right) - \mu^2 - \sigma^2\right) \quad (2)$$

Eq.(2) is used to calculate the summation of each dimension in the potential space. For example, if there are 3 dimensions in the latent space, $(1 + \log(\sigma^2) - \mu^2 - \sigma^2)$ will be calculated three times in each dimension. The final loss function is defined as:

$$\text{Loss}_{SWSRM} = \alpha \cdot RMSE + D_{KL} \qquad (3)$$

where $\alpha$ is the reconstructed loss weight, it can be set empirically.

## 3.2 Spectrogram Super-Resolution Module

**Module Architecture.** As shown in Figure 3, the SSRM is a U-Net-based multi-scale dual regression network. The network consists of two parts: feature extraction and spectrogram reconstruction [10]. The orange lines show the additional supervision of the double regression. Among them, the multi-scale residual network (MSRN) is composed of the multi-scale residual blocks [18]. MSRN is used as the feature extraction. Residual blocks are used to extract multi-scale features of each scale spectrogram. An additional constraint is simultaneously introduced to reduce the possible space, allowing a more accurate reconstruction of the SR spectra. If the mapping from LR to HR is optimal, the SR spectrogram can be downsampled to obtain the same input LR spectrogram. The dual regression process does not depend on the HR spectrogram and can be learned directly from the LR spectrogram.

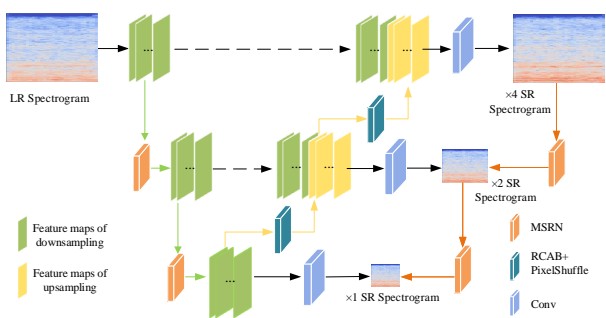

**Figure 3: Architecture of spectrogram super-resolution reconstruction networks.**

The LR spectrogram is first used as input in the feature extraction. Then the convolutional layers and MSRN with various size of convolutional kernels detect the feature maps at different scales adaptively. The features information is shared by jump connections at different scales. The residual structure is used to extract more detailed features. And the reconstruction module takes the feature maps of the downsampling as input.

We used the residual channel attention block (RCAB) [42] and PixelShuffle [34] in the spectrogram reconstruction. By using a channel attention mechanism, the features of each channel are adaptively rescaled by the interdependencies between feature channels. During upsampling, PixelShuffle can reorganize the low-resolution feature maps between multiple channels of the convolution kernel to obtain high-resolution feature maps.

**Loss.** Assume $H(x)$ is a high-resolution spectrogram, the downsampled spectrogram $D(H(x))$ should be very close to the input low-resolution spectrogram $x$. Given a set of $N$ paired samples $S_H = \{(x_1, y_1), (x_2, y_2), ..., (x_N, y_N)\}$, where $x_i$ and $y_i$ denote the

$i^{th}$ pair of low and high-resolution spectrogram in $S_H$. The loss defined as:

$$\text{Loss}_{SSRM} = \sum_{i=1}^{N} L_H\left(H\left(x_i\right), y_i\right) + \lambda L_D\left(D\left(H\left(x_i\right)\right), x_i\right) \quad (4)$$

where $L_H$ and $L_D$ denote the $L_1$ loss of original U-Net and dual regression network, respectively. Here, $\lambda$ controls the weights of $L_D$.

The $L_1$ loss is calculated as the sum of absolute differences between the HR spectrogram and SR spectrogram. $L_1$ loss makes the generated SR spectrogram as close as possible to the HR spectrogram. It can be defined as:

$$L_1\left(p, \hat{p}\right) = \sum_{i=0}^{m}\left|p^{(i)} - \hat{p}^{(i)}\right| \quad (5)$$

where $p$ and $\hat{p}$ denote the pixel value of the HR spectrogram and SR spectrogram at same coordinate point, respectively. In the experiments, only dual regression loss $L_D$ is added to LR spectrogram, which can reduce the calculate computation while improving the performance.

## 3.3 Time-Frequency Domain Fusion Module

Referencing [40], we propose an attention-based time-frequency domain fusion module (TFDFM) to combine the dual-domain SR results. The TFDFM uses attentive feature pyramid module composed of multiple pyramid units to acquire multi-level dual-domain features. In each pyramid unit, fixed-size multi-scale attention blocks capture intra- and inter-modality interaction information. To combine the pyramid units, an adaptive fusion layer is utilized. This layer integrates the pyramid units in a selective fusion manner by exploring the correlation between multi-level features.

The architecture of the pyramid time-frequency domain fusion layer is shown in Figure 4. The features extracted from the SWSRM and the SSRM are as input of TFDFM. In each pyramid unit, intra- and inter-modality interactions are first introduced using a fixed-size attention mechanism, features are integrated through an extended residual convolution block. Finally, the outputs of all units are saved as pyramid-like multi-modal features. The final fusion result is obtained by combining the pyramid features through the adaptive fusion module.

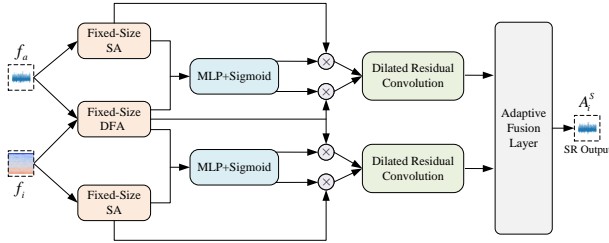

**Figure 4: Time-Frequency Domain Fusion Module.**

In the TFDFM, the self-attention (SA) and dual-domain fusion attention (DFA) are used to provide the interaction of temporal features. The attention score between different audio clips is calculated by scaling the dot-product attention, as shown in Eq. (6).

$$\text{Attention}\left(q, k, v\right) = \text{Softmax}\left(\frac{qk^T}{\sqrt{d_m}}\right)v \quad (6)$$

where $q, k, v$ denote the the *query*, *keyword*, and *value vectors*, respectively, $d_m$ is the dimension of the *query vector*, and $T$ is the matrix transpose operation.

The SA block is defined as:

$$\text{SA}\left(f\right) = \text{Attention}\left(fW_q, fW_k, fW_v\right) \quad (7)$$

where $W_q, W_k, W_v$ is the learnable parameters and $f$ is the input feature.

For DFA blocks, the features in the current modality are assigned as *query vector*, while the *keyword vector* and *value vector* are derived from features in other modalities, as shown in Eqs. (8) and (9).

$$\text{DFA}\left(f_a, f_i\right) = \text{Attention}\left(f_aW_q, f_iW_k, f_iW_v\right) \quad (8)$$
$$\text{DFA}\left(f_i, f_a\right) = \text{Attention}\left(f_iW_q, f_aW_k, f_aW_v\right) \quad (9)$$

where $f_a$ is the audio feature and $f_i$ is the spectral feature.

The DFA blocks share the parameter matrix that can project audio and spectral features into the same subspace and further combines the multi-modal features. Subsequently, the feedforward layer processes the features. The TFDFM uses the normalizing layer for regularization and uses the residual connection for identity mapping to avoid overfitting. The interaction window is utilized to limit the interaction size between SA and DFA layers.

To limit interaction windows, some masks are added in the irrelevant regions of signals. The fixed-size attention is calculated by

$$\text{SA}\left(f, d\right) = \text{Attention}\left(fW_q, S_t\left(f, d\right)W_k, S_t\left(f, d\right)W_v\right) \quad (10)$$

$$\text{DFA}\left(f_i, f_a, d\right) = \text{Attention}\left(f_iW_q, S_t\left(f_a, d\right)W_k, S_t\left(f_a, d\right)W_v\right) \quad (11)$$

$$\text{DFA}\left(f_a, f_i, d\right) = \text{Attention}\left(f_aW_q, S_t\left(f_i, d\right)W_k, S_t\left(f_i, d\right)W_v\right) \quad (12)$$

$$S_t\left(x, d\right) = \left[x_{t-d}, ..., x_{t+d}\right] \quad (13)$$

where $S_t$ is the created interaction windows for the $t^{th}$ data, $d$ is the size of the window.

The outputs of SA and DFA are first connected along the channel dimension. Then, the refined features of channel attention are computed by linear layers and S-shaped functions. Finally, the SR audio is combined by summing the refined single-modal and double-modal features.

# 4 EXPERIMENTAL RESULTS

## 4.1 Dataset

We perform the TFDFE-ASR experiments and analyse the experimental results on the VCTK dataset [37], Piano dataset [27], and ShipsEar dataset [33]. The VCTK dataset contains speech data from 108 English speakers. Each speaker reads about 400 different sentences, for a total of 44 hours. The Piano dataset contains 10 hours of Beethoven sonata with a sampling rate of 44.1kHz. Considering that the audio can propagate in water, we adopt an underwater acoustic dataset, ShipsEar, to test the performance of the TFDFE-ASR method. The ShipsEar includes the ship radiated noise data recorded in different regions of the Spanish coast from 2012 to 2013. The dataset consists of 90 acoustic records of 11 types of ships and

environmental noise within 15 seconds to 10 minutes. According to the annotations in the original dataset, the ships can be divided into four categories, namely A, B, C, and D, as well as E of environmental noise.

## 4.2 Parameters of Experimental System

The proposed method is performed on Tensorflow with an NVIDIA TITAN XP graphics card and CUDA version 11.0. The SWSRM is set as follows: the number of iterations during the module training period is 150, where a random gradient descent optimization model is used. The learning rate was initially set at $10^{-5}$, with a batch size of 32. The network input of the SSRM is obtained by downsampling the original data in the dataset through Bicubic interpolation. The original data is the spectral map obtained by Fourier transform of the audio dataset, and then the image super-resolution downsampling method is used to produce ×2, ×4, and ×8 times of the paired spectral dataset. The SSRM is set as follows: we apply Adam with $\beta_1 = 0.9$, $\beta_2 = 0.999$ and set minibatch size as 32. The learning rate is initialized to $10^{-4}$. Through the TFDFM, the feature information from the SWSRM and the SSRM is extracted and fused to output the final super-resolution audio signal.

## 4.3 Evaluation Indexes

Three objective and subjective evaluation indexes are employed to measure the quality of SR audio signals by comparing them with actual HR audio. The objective indexes include the signal-to-noise ratio (SNR) and the logarithmic spectral distance (LSD) [9]. A higher SNR indicates better audio quality, while a lower LSD indicates better audio quality. The mean opinion score (MOS) is employed as the subjective evaluation index, which is divided into five levels from high to low. The higher the score, the better the audio quality. By grading the same sentence differently, 50 evaluators are required to rate multiple sounds and obtain the final average score.

## 4.4 SR Experiments on Audio Signals in the Air

We first test the SR quality of audio signals propagated in the air, including the VCTK dataset and the Piano dataset. Each audio file of the VCTK is 16-bit, and a mono voice file with a sampling rate of 48kHz is selected for the experiment. The VCTK includes 223 audio files of single speaker (VCTK$_S$) and 5878 audio files of multi speakers (VCTK$_M$). The dataset was first divided according to 88% training set, 6% validation set, and 6% test set. Then the original data is downsampled to 16kHz, the target data is downsampled to 8kHz, and the resulting signal is interpolated with three splines to obtain a differentiated speech signal. The original audio is converted into a logarithmic spectrogram by Fourier transform, and a total of 800 spectrograms are randomly selected as the training dataset in the SSRM. Arbitrarily selecting 100 spectrograms as the validation set and 5 spectrograms as the test set. And trained for a total of 500 iteration cycles.

Each audio file of the Piano dataset with sampling rate of 44.1kHz. 10 hours of data are approximately divided into a series of 12 second audios, totaling 2868 files. The data is divided into a training set (88%), a validation set (6%), and a testing set (6%). The data for SSRM undergoes the same preprocessing as the VCTK dataset task.

The traditional cubic spline interpolation is used as a comparison method of the audio SR to upsample the LR audio signal [8]. The first comparative method based on deep learning audio SR is SFSR-Net [32]. It can separate the mixed audio signals, and reconstruct the missing information in the upper frequencies by operating on the spectrograms of the output audio source estimations. The second comparative method is RFD-Net [14], which is a recursive structure to iteratively refine and extract hierarchical audio feature. It employs an up-and-down sampling learner, and captures the deep relationships between HR and LR audio pairs, thus producing high-quality audio. The third comparative method is TFiLM [3], which uses recurrent neural networks to modify the activation of convolutions.The fourth comparative method is NU-Wave [17] which is a diffusion probabilistic model based on neural vocoders. The fifth comparative method is NU-Wave 2 [11]. It is a diffusion model for neural audio upsampling that enables the generation of 48kHz audio signals from inputs of various sampling rates with a single model. As shown in the data on the left side of the comparison section of Table1, The TFDFE-ASR compares SNR and LSD with other audio SR models on VCTK$_S$, VCTK$_M$, and Piano datasets, respectively. $R$ denotes the upscaling factor, $R = 2$ denotes upsampling from 8kHz to 16kHz, $R = 4$ denotes upsampling from 4kHz to 16kHz and $R = 6$ denotes upsampling from 4kHz to 24kHz.

**Table 1: Objective evaluation comparing with other methods.**

| Model | R | VCTK$_S$ | | VCTK$_M$ | | Piano | | ShipsEar | |
|---|---|---|---|---|---|---|---|---|---|
| | | SNR | LSD | SNR | LSD | SNR | LSD | SNR | LSD |
| Spline | 2 | 19.07 | 1.99 | 18.89 | 2.08 | 15.48 | 2.27 | 16.21 | 3.25 |
| SFSRNet | 2 | 20.82 | 1.36 | 19.94 | 1.32 | 25.35 | 2.07 | 18.42 | 1.34 |
| RFD-Net | 2 | 21.11 | 1.24 | 19.84 | 1.34 | 24.71 | 2.15 | 18.47 | 1.44 |
| TFiLM | 2 | 20.34 | 1.55 | 19.81 | 1.82 | 25.42 | 2.01 | 18.83 | 1.59 |
| NU-Wave | 2 | 19.93 | 1.76 | 19.35 | 1.51 | 24.69 | 2.27 | 18.33 | 1.52 |
| NU-Wave 2 | 2 | 21.78 | 1.17 | 20.24 | 1.45 | 25.28 | 2.11 | 19.55 | 1.63 |
| TFDFE-ASR | 2 | **21.83** | **1.16** | **20.74** | **1.25** | **25.67** | **1.94** | **22.16** | **1.26** |
| Spline | 4 | 15.33 | 3.13 | 13.42 | 2.99 | 12.43 | 2.23 | 11.84 | 4.50 |
| SFSRNet | 4 | 17.29 | 1.41 | 16.65 | 1.40 | 18.81 | 2.32 | 17.63 | 2.56 |
| RFD-Net | 4 | 18.35 | 1.33 | 17.32 | 1.22 | 18.62 | 2.25 | 17.16 | 2.97 |
| TFiLM | 4 | 17.88 | 2.15 | 16.47 | 1.75 | **19.33** | 2.26 | 16.88 | 3.41 |
| NU-Wave | 4 | 17.24 | 1.31 | 16.42 | 1.59 | 17.53 | 2.21 | 16.56 | 3.01 |
| NU-Wave 2 | 4 | 18.63 | 1.46 | 17.66 | 1.39 | 18.86 | 2.34 | 17.94 | 2.14 |
| TFDFE-ASR | 4 | **19.83** | **1.29** | **18.33** | **1.15** | 18.95 | **2.14** | **18.06** | **2.08** |
| Spline | 6 | 12.29 | 6.94 | 9.88 | 6.84 | 10.76 | 4.06 | 8.64 | 7.11 |
| SFSRNet | 6 | 15.36 | 3.73 | 15.83 | 4.52 | 14.12 | 3.22 | 14.06 | 3.82 |
| RFD-Net | 6 | 12.63 | 4.21 | 15.22 | 4.69 | 14.33 | 3.13 | 13.11 | 4.52 |
| TFiLM | 6 | 12.95 | 4.38 | 12.03 | 3.94 | 13.36 | 3.86 | 12.86 | 4.43 |
| NU-Wave | 6 | 12.23 | 4.16 | 14.65 | 3.57 | 13.48 | 3.45 | 12.91 | 4.55 |
| NU-Wave 2 | 6 | **16.41** | 3.89 | 16.51 | 3.62 | 15.96 | 3.19 | 15.12 | 3.36 |
| TFDFE-ASR | 6 | 15.69 | **3.26** | **16.56** | 3.43 | **16.12** | **3.14** | **15.23** | **3.24** |

As can be seen from Table 1, the SNR of TFDFE-ASR audio on the VCTK dataset with $R = 2$ and $R = 4$ has higher values compared with the SR results of other methods. An obvious decline is observed in the LSD values. On the Piano dataset, although the SNR value is not the best among the comparison models when $R = 4$, the experimental result of TFDFE-ASR is satisfactory. Proving that the TFDFE-ASR not only performs well on the VCTK dataset but also

on non-vocal music signals. In addition, TFDFE-ASR performed well in single speaker task with $R = 6$, while in multi speakers and Piano tasks, TFDFE-ASR did not perform significantly compared to other comparison methods.

A comparison was made with other experimental models on the subjective evaluation index MOS. MOS is divided into five levels from 5 to 1, and a higher score indicates better speech quality [1]. The recording was randomly selected from the results of six methods, as well as a clean recording of the original version. 200 workers participated in this experiment, and we collected a total of 14000 ratings. Finally, the average score of all evaluators is calculated. The MOS score of ground-truth in the VCTK dataset is 4.31, while the MOS score of ground-truth in the Piano dataset is 4.59. The evaluation scores of MOS are shown in Figure 5, from which it can be seen that TFDFE-ASR works significantly on the non-vocal Piano dataset. It proves that the model not only has good results on the speaker dataset but also can achieve good results for non-vocal audio signals like music.

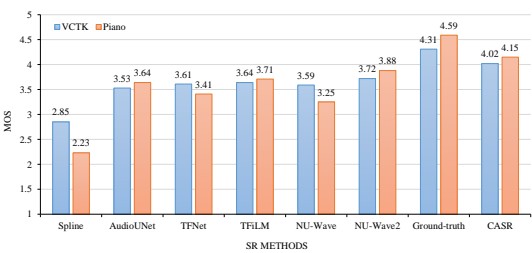

**Figure 5: Comparison results of subjective evaluation criteria.**

The spectrograms after audio super-resolution are shown in Figure 6. Fig. 6(a) displays the spectrograms of the LR audio signal, Fig. 6(b) corresponds to the spectrum of the original HR audio signal, and Fig. 6(c) presents the spectrograms of the SR audio signal reconstructed by the TFDFE-ASR. The first row shows the spectrograms of an audio segment in the VCTK. The second row is the spectrograms of a piece of music on the Piano. It can be seen that the TFDFE-ASR method can reconstruct the missing high-frequency components well in the SR results.

## 4.5 SR Experiments of Underwater Acoustic Signals

The comparison results between TFDFE-ASR and other audio SR models on the ship hydroacoustic signal data set SNR and LSD are shown in the data on the right side of table 1. From the experimental results in Table 1, it can be seen that TFDFE-ASR outperforms the comparative models in the ShipsEar dataset.

The ShipsEar dataset is sliced into audio data of 1 second, where categories A, B, C, D, and E have 1875, 1560, 4270, 2455 and 1140 audio segments, respectively. On the ShipsEar dataset, we measure and analyze the impact of the TFDFE-ASR on the target recognition accuracy of underwater acoustic signals. The task of underwater acoustic target recognition adopts the method of [13]. Input the extracted fusion features into a model constructed by an 18 layer

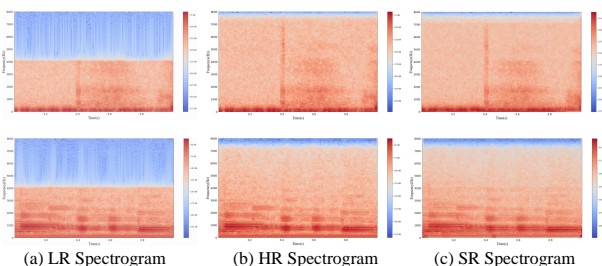

(a) LR Spectrogram   (b) HR Spectrogram   (c) SR Spectrogram

**Figure 6: Super-resolution spectrograms on the VCTK, Piano datasets.**

residual network for underwater acoustic signal target recognition experiments. We verify that the results of audio SR can better improve the accuracy of target recognition in the experiments.

Figure 7 shows the time domain waveforms and Mel spectrogram of the original ShipsEar data. The up and bottom rows respectively show the time domain waveforms and Mel spectrograms of the original audio and the corresponding audio SR signals of the five types of ships. Each sample has a duration of 1 second and a sampling rate of 16kHz. Comparing the two spectrograms above and below, we can see that the SR results have more details in sound waves and spectrograms than the original data.

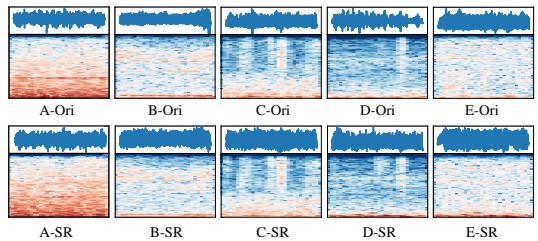

A-Ori   B-Ori   C-Ori   D-Ori   E-Ori

A-SR   B-SR   C-SR   D-SR   E-SR

**Figure 7: Comparison of time and frequency domains of the original audio and super-resolution audio on the ShipsEar dataset.**

The *Accuracy*, *Precision*, *Recall*, and *F1-score* are employed as the evaluation indexes of underwater target recognition performance. The detailed experimental results of each category are shown in Table 2. Support in the table indicates the number of samples of this category in the test. Compared to the original audio, the accuracy of target recognition has improved after using SR audio. The target recognition experiments with super-resolution performed better on all categories of data, except for the category E environmental noise, which had slightly lower indicators than the original data. The model increased the target identification by about 12.66%, from 84.55% on the original data to 97.21% on the generated data. The SR results of Spline, SFSRNet, RFD-Net, and TFiLM on the ShipsEar dataset have almost unchanged from the original signals.

## 4.6 Ablation Experiment

In the ablation experiments, we respectively test and analyze the impact of SWSRM, SSRM, and TFDFM for the proposed TFDFE-ASR. By combining experimental methods, we analyze whether the

**Table 2: Target recognition experiment results(%) on the ShipsEar dataset.**

| Category | Precision | | Recall | | F1-score | | Support |
|---|---|---|---|---|---|---|---|
| | Ori | SR | Ori | SR | Ori | SR | |
| A | 82.70 | 98.72 | 82.70 | 98.22 | 82.70 | 98.45 | 393 |
| B | 80.71 | 91.96 | 77.47 | 91.37 | 79.06 | 91.67 | 311 |
| C | 85.41 | 96.79 | 87.91 | 97.37 | 86.64 | 97.09 | 843 |
| D | 85.86 | 100.00 | 86.73 | 99.80 | 86.91 | 99.90 | 495 |
| E | 87.16 | 97.24 | 82.97 | 97.25 | 84.63 | 97.25 | 218 |

Target recognition *Accuracy* on original acoustic signals: 84.55%
Target recognition *Accuracy* on SR acoustic signals: 97.21%

information of sound waves and spectrograms has been effectively utilized, and verify the role of the proposed modules in the audio super-resolution effect. The results of the ablation experiment are shown in Table 3. By conducting experiments on each module on three datasets, the impact of each module on model performance was evaluated. Table 3 describes the SNR and LSD of SR results by SWSRM, SSRM, and TFDFE-ASR on the above three datasets. It can be found that the SWSRM performs better on audio signals in the air (VCTK dataset and Piano dataset). The SSRM has a greater impact on underwater acoustic signals (ShipsEar dataset).

**Table 3: The impact of each module on model performance.**

| Module | VCTK$_S$ | | Piano | | ShipsEar | |
|---|---|---|---|---|---|---|
| | SNR | LSD | SNR | LSD | SNR | LSD |
| SWSRM | 17.53 | 1.89 | 22.07 | 1.36 | 16.89 | 2.08 |
| SSRM | 15.32 | 2.16 | 20.82 | 1.99 | 19.94 | 1.62 |
| TFDFE-ASR | 21.83 | 1.16 | 25.67 | 1.94 | 22.16 | 1.26 |

In addition, we also perform target identification on the SR results (sound wave and spectrogram) of the ShipsEar dataset, respectively. The method of [19] was applied to the spectrogram for target recognition of acoustic signals using a neural network model with a self-attentive mechanism. In Table 4, it can be seen that the SR spectrogram outperformed the SR sound wave in terms of target recognition accuracy. Various categories of targets show that the TFDFE-ASR achieves a more prominent effect than only conducting a single module (SWSRM or SSRM). The TFDFE-ASR algorithm can make more contributions to target recognition.

**Table 4: The impact of each module on the accuracy(%) of underwater acoustic signal target recognition.**

| Category | SWSRM (Sound wave) | SSRM(Spectrogram) | TFDFE-ASR |
|---|---|---|---|
| A | 86.76 | 94.40 | 98.72 |
| B | 84.24 | 89.38 | 91.96 |
| C | 88.02 | 93.95 | 96.79 |
| D | 93.33 | 96.97 | 100.00 |
| E | 90.37 | 92.20 | 97.24 |
| Avg. | 88.67 | 93.89 | 97.21 |

Through the above comparative and ablation experiments, we found that different modules play different roles on the three datasets. The underwater acoustic signals are mainly characterized by low frequencies compared to audio signals in air, which are mainly dominated by high frequencies. SWSRM has a good super-resolution effect on high-frequency signals, while SSRM has a good super-resolution effect on low-frequency signals. By counting the fusion effect of these three datasets with a total of about 20,000 audio samples, we found that the fusion weights $w_{SWSRM}$ and $w_{SSRM}$ for audio SR closely resemble the weighting relationship of a hyperbolic tangent function. Their functional form is shown in the formula:

$$\begin{cases} w_{SWSRM} = \frac{1+\tanh(\omega-\bar{\omega})}{2} \\ \\ w_{SSRM} = \frac{1-\tanh(\omega-\bar{\omega})}{2} \end{cases} \quad (14)$$

where $\omega$ is the frequency of the acoustic signal, $\bar{\omega}$ representing the mean of the signal frequency in the dataset. Therefore, in practical applications, the fusion module can be defined as an activation function based on Eq. (14) for the convenience of calculation. The curve relationship of the function is shown in Figure 8.

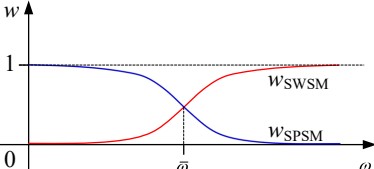

**Figure 8: Weight relationship curves of SWSRM and SSRM in TFDFM.**

## 5 CONCLUSION

In this paper, we propose an time-frequency domain fusion audio super-resolution method that fully learns the correlation between the time and frequency domains. The time-frequency domain fusion approach preserves the commonality and uniqueness of the sound wave signal in the frequency domain and time domain to the greatest extent, improves the quality of the audio signal, and makes the audio clearer. The method consists of three modules: VAE-based SWSRM, U-Net-based SSRM and attention-based TFDFM. Among them, SWSRM is an improved 1D-VAE that can generate more high-frequency components for audio. At the same time, SSRM enhances the details (texture) of the LR spectrum, especially in the low-frequency component area. Finally, self-attention and dual-domain fusion attention are used in TFDFM to better realize time-frequency domain content interaction. The proposed method achieves excellent audio super-resolution results. Experiments on VCTK and Piano datasets demonstrate the effectiveness of audio super-resolution tasks. Meanwhile, the target recognition results on the ShipsEar dataset show that applying our method to hydroacoustic data can significantly improve the quality of hydroacoustic signals and the accuracy of target recognition.

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
