# OpenReview forum: "Time-Frequency Domain Fusion Enhancement for Audio Super-Resolution"
_acmmm.org/ACMMM/2024/Conference — MM2024 Poster_

### Official Review · Reviewer_o4uC · 2024-04-29

**Rating:** 2
**Confidence:** 3

**Summary:**

In this paper, the authors propose a time-frequency domain fusion audio super-resolution method that fully learns the correlation between the time and frequency domains. This approach preserves the commonality and uniqueness of the sound wave signal in both the frequency and time domains to the greatest extent possible, thereby enhancing the quality of the audio signal and making it clearer.

**Strengths:**

1. The experimental section of the paper is relatively comprehensive, with the authors designing experiments to validate the effectiveness of their method for downstream tasks.
2. The textual descriptions and graphical illustrations of the method in the paper are clear and coherent.

**Limitations:**

1. This work “Time-Frequency Networks for Audio Super-Resolution” has already achieved audio super-resolution through the fusion of time-frequency information. TFDFE-ASR is an incremental work and does not possess significant novelty.
2. There is an error in Table 1. When R = 6, RFD-Net exhibits the best performance on the LSD metric on the Piano dataset.
3. Figure 6 is insufficient to demonstrate the advantage of TFDFE-ASR in reconstructing high-frequency components compared to other methods. Additionally, the font size in Figure 6 is too small to read clearly.
4. "The SR results of Spline, SFSRNet, RFD-Net, and TFiLM on the ShipsEar dataset have almost unchanged from the original signals. (Line 804-806)" This statement would benefit from supporting data.
5. How do you prove that "SWSRM has a good super-resolution effect on high-frequency signals, while SSRM has a good super-resolution effect on low-frequency signals. (Line 875-877)"?
6. The comparison with methods from 2023 is missing.

**Suitability:**

3

---

### Official Review · Reviewer_5xMB · 2024-05-23

**Rating:** 4
**Confidence:** 1

**Summary:**

This paper presents a time-frequency domain fusion audio super-resolution method aimed at enhancing the quality of acoustic signals. The proposed method integrates a variational autoencoder-based sound wave super-resolution module, a U-Net-based spectrogram super-resolution module, and an attention-based time-frequency domain fusion module. Experiments on the different datasets demonstrate the state-of-the-art sr performance.

**Strengths:**

1. The authors introduce a novel approach that leverages both time and frequency domain information for audio super-resolution, capturing a more comprehensive audio features.
2.  The method achieves state-of-the-art results on the VCTK and Piano datasets, indicating its effectiveness for both vocal and non-vocal audio signals.

**Limitations:**

1. The paper employs a Variational Autoencoder (VAE) to enhance the time-domain audio, which is unconventional as VAEs are traditionally used for data compression rather than enhancement in vision tasks.
﻿
2. The details of the ablation study are not explicitly clear in the paper. It is not evident whether the removal of one domain module was conducted to eliminate the dual-domain interaction or if it was replaced with an alternative module.

**Suitability:**

2

---

### Official Review · Reviewer_Dmpu · 2024-05-24

**Rating:** 4
**Confidence:** 2

**Summary:**

This paper proposes an end-to-end time-frequency domain fusion enhanced audio super-resolution network, which consists of three main modules, including  sound wave super-resolution module (SWSRM), U-Net-based Spectrogram Super-Resolution Module (SSRM), and attention-based Time-Frequency Domain Fusion Module (TFDFM). Then a weighted fusion is applied to obtain a super-resolution audio signal.

**Strengths:**

1.	Propose a network that obtains both the sound wave super-resolution in time domain and spectrogram super-resolution in frequency domain, then fuse them to get the super-resolution audio. The proposed network is presented in detail.
2.	Three kinds of experiments are conducted on speech, piano and underwater acoustic datasets. The results are promising.

**Limitations:**

1.	Although this article proposes a network that fuses audio's time domain signal and spectrum domain signal, its network structure only combines the commonly used CNN network and UNet network, and similar structures are also common in other applications. From this perspective, it is not very innovative.
2.	The proposed method uses the information of both time and frequency domains that extract richer features, but the compared methods do not include time-frequency based methods, such as references [22] and [35].
3.	The real-world audio applications of high-fidelity usually have the sampling rate greater than or equal to 44 kHz. In the proposed method, the sampling rates tested are between 4kHz to 24kHz. No verification experiments have been done for R=8 or higher.

**Suitability:**

2

---

### Official Review · Reviewer_wNbQ · 2024-05-25

**Rating:** 4
**Confidence:** 1

**Summary:**

This paper proposes a two-branch framework for the audio super-resolution task. One branch works in the sound wave space using VAE, and the other branch works in the spectrogram space using U-Net. The two branches are merged using a fusion module.

**Strengths:**

1. The proposed framework is intuitive.

2. The writing is good in general with good presentations and illustrations.

3. The experiments show that the proposed method is effective compared to some important prior works.

**Limitations:**

1. In the experiments, the authors should include more 'recent' methods that can fully represent the state-of-the-art.

2. It will be better to explain the figures in the captions. Current captions are not very informative.

3. It will be better to avoid using too much abbreviations, such as SWSRM, SSRM, and TFDFM. There are too much abbreviations and they are similar, lowering the readability.

**Suitability:**

2

---

### Meta-Review · Area_Chair_rMiB · 2024-06-30

**Recommendation:** Accept (Poster)
**Confidence:** 3

**Metareview:**

Initially the paper received 3 borderline accept and 1 weak reject. Through the rebuttal, 1 weak reject was changed to borderline accept. The reviewers acknowledged that the paper is well written and the proposed method performs well. Overall, the paper received all positive ratings.